# Room Temperature Operation of UV Photocatalytic Functionalized AlGaN/GaN Heterostructure Hydrogen Sensor

**DOI:** 10.3390/nano11061422

**Published:** 2021-05-28

**Authors:** June-Heang Choi, Taehyun Park, Jaehyun Hur, Ho-Young Cha

**Affiliations:** 1School of Electronic and Electrical Engineering, Hongik University, Seoul 04066, Korea; petrus0314@mail.hongik.ac.kr; 2Department of Chemical and Biological Engineering, Gachon University, Seongnam 13120, Gyeonggi, Korea; thpark@gachon.ac.kr

**Keywords:** AlGaN/GaN, ZnO-nanoparticles, Pd, photocatalyst, hydrogen sensor, ultraviolet

## Abstract

An AlGaN/GaN heterostructure based hydrogen sensor was fabricated using a dual catalyst layer with ZnO-nanoparticles (NPs) atop of Pd catalyst film. The ZnO-NPs were synthesized to have an average diameter of ~10 nm and spin coated on the Pd catalyst layer. Unlike the conventional catalytic reaction, the fabricated sensors exhibited room temperature operation without heating owing to the photocatalytic reaction of the ZnO-NPs with ultraviolet illumination at 280 nm. A sensing response of 25% was achieved for a hydrogen concentration of 4% at room temperature with fast response and recovery times; a response time of 8 s and a recovery time of 11 s.

## 1. Introduction

Hydrogen is presently being studied as an alternative energy source for eco-friendly renewable energy generation instead of fossil fuels [1,2]. The successful utilization of hydrogen requires highly sensitive and reliable safety sensors. For example, hydrogen explosions occur at concentrations exceeding 4.65% [3,4,5]; therefore, the sensor must detect low levels of hydrogen quickly and accurately.

Among various types of sensors, an FET-type sensor with a catalytic gate material has been widely studied due to its compact size and low power consumption. A catalytic reaction with a target gas changes the surface potential of FET, which in turn modulates the FET current [5,6,7].

Gallium nitride (GaN) has a wide energy bandgap of 3.4 eV [8,9,10,11,12,13], which results in a low intrinsic carrier density allowing GaN to maintain its semiconductor properties at much temperatures higher than those possible with Si or GaAs. When an AlGaN/GaN heterostructure is formed, a 2-dimensional electron gas (2DEG) channel is created with high mobility at the interface [14,15,16]. Since the AlGaN thickness is generally of the order of a few tens of nanometers, the channel current strongly depends on the surface potential change, thus resulting in sensitive responses from the sensor. Therefore, an AlGaN/GaN heterostructure would be a great candidate for the sensing platform.

Various catalysts have been investigated for use in hydrogen sensors, including metals and metal oxides. Pt, Pd, and Ru are known to have high hydrogen solubility [17,18,19,20,21,22,23,24,25,26]. Metal oxides, such as ZnO, TiO_2_, SnO_2_, WO_2_, PdO, In_2_O_3_, and Fe_2_O_3_, have been studied mostly as resistive sensors where adsorbed oxygen ions play an important role in the reaction mechanism [27,28,29,30,31,32,33,34,35,36,37,38,39,40].

Since the typical catalytic reaction with hydrogen occurs at elevated temperatures [41], the sensor module must be heated using a separate or integrated heater, which incurs extra power and time for stabilization. In addition, thermal heating of the catalyst and sensor platform degrades long-term reliability [42]. Therefore, there is a strong demand for a hydrogen sensor operating at room temperature without heating. A possible solution would be a photocatalytic reaction where the gas reaction with catalyst material is accelerated by photoreaction. Indeed, it was reported that metal oxides reacted with hydrogen at room temperature with exposure to ultraviolet (UV) light [43,44,45,46,47]. In this study, an AlGaN/GaN FET sensor was fabricated using a dual catalyst layer with ZnO-nanoparticles (NPs) atop of Pd catalyst film, which exhibited room temperature sensing capability of hydrogen with UV illumination.

## 2. Experiment and Results

Figure 1 illustrates the synthesis process of ZnO-NPs. The precursor solution was the hydrolysis of Zn(CH_3_COO)_2_·2H_2_O (Sigma-Aldrich, St. Louis, MO, USA) with KOH (Samchun, Samchun, Seoul, South Korea, 95%) in methanol (Sigma-Aldrich, St. Louis, MO, USA, 99.9%). ZnO-NPs formed after 2 h reaction were separated by centrifugation. A detailed synthesis process and the characterization for the synthesized ZnO-NPs can be found in ref [48].

The crystal structure of the ZnO-NPs was investigated using powder X-ray diffraction, which is shown in Figure 2a. The crystal planes of the crystalline ZnO-NPs corresponding to the observed diffraction peaks are indicated in the plot, confirming the successful synthesis of ZnO-NPs. The internal and surface chemical bonding state of ZnO-NPs was analyzed through X-ray photoelectron spectroscopy (XPS). Figure 2b represents the O 1 s spectrum of ZnO-NPs where three distinct binding energy peaks were observed at 529.6, 531.0, and 532.1 eV, which corresponded to the O atom in the Zn-O bonding (lattice oxygen), oxygen sublattice imperfection (oxygen vacancy) and surface adsorbed O_2_^−^ of ZnO-NPs, respectively [49,50,51]. This XPS result demonstrates the presence of the ionized oxygen molecules at the ZnO-NP surface, which plays an important role in the catalytic reaction under UV illumination. The absorption spectrum of ZnO-NPs as a function of wavelength was also measured to estimate the optical bandgap of ZnO-NPs. Figure 3a,b show the absorption spectrum and Tauc plot of the ZnO-NPs thin film from which the optical bandgap was estimated to be 3.24 eV. The wide band gap of ZnO-NPs can selectively absorb UV light that can remove the oxygen molecules adsorbed on the ZnO-NPs surface.

Figure 4a,b shows Transmission Electron Microscopy (TEM) and High-Resolution TEM (HRTEM) images of ZnO-NPs, respectively. The size of ZnO-NPs was in the range of 5–10 nm. The lattice fringe spacings of 0.28 and 0.25 nm observed in HRTEM image are related to (100) and (101) planes of the ZnO-NPs, respectively. Other crystalline planes such as (102) and (110) were also confirmed from the Fast Fourier transform (FFT) pattern shown in the inset of Figure 4b.

The AlGaN/GaN epitaxial structure used for sensor fabrication comprised a 10 nm in situ SiN_x_ passivation layer, a 3.5 nm GaN layer, a 23 nm Al_0.24_Ga_0.76_N layer, and a 4.2 µm GaN layer on a Si (111) substrate. After defining the ohmic contact region, the exposed in-situ SiN_x_ layer was etched using SF_6_-based inductively coupled plasma reactive ion etching (ICP-RIE) and the underlying GaN and AlGaN layers were etched down to the middle of the AlGaN layer using Cl_2_/BCl_2_-based ICP-RIE. A Ti/Al/Ni/Au metal stack was evaporated as the ohmic contact metal and a rapid thermal annealing process was carried out at 830 °C for 30 s in an N_2_ ambient. Then, device isolation was performed using the same etching process used for the ohmic contact formation, with a larger etch depth of 350 nm. The pad electrodes were formed using a Ti/Au metal stack. Then, the catalyst area was defined by photolithography. In order to lower the standby current to achieve high sensitivity [52], the catalyst area was also recessed using the same plasma etching method used for the ohmic contact process. The final thickness of the AlGaN layer after being etched was 10 nm. After evaporating a 30 nm Pd catalyst layer, the sensor surface was passivated with a 100 nm SiN_x_ film. The passivation film on the catalyst region and pad contact area were etched using a SF_6_-based ICP-RIE. The synthesized ZnO-NPs were spin coated on the Pd layer after being dispersed in chloroform/ethanol solution using an ultrasonicator for 2 h. Finally, the sample was annealed at 120 °C for 1 h. The thicknesses of the ZnO-NP layer was 170 nm. The fabricated sensor with the ZnO-NP/Pd dual catalyst layer is illustrated in Figure 5, where the inset is the cross-sectional TEM image of the dual catalyst layer.

Sensor characteristics were measured at room temperature with and without UV illumination, where the UV light source was a 280 nm LED operated by a driving current of 180 mA resulting in an optical power density of 1.82 W/cm^2^ at the catalyst surface. The hydrogen concentration used for the tests was 4%.

The current–voltage characteristics without and with UV illumination at 280 nm are compared in Figure 6a,b, respectively. While little changes were observed in the sensor current under hydrogen injection without UV illumination, a significant increase was observed under hydrogen injection with UV illumination. The sensing mechanism is illustrated in Figure 7. There are negatively ionized oxygen species (O_2_^−^(ad)) adsorbed at the ZnO surface in air, which have strong adhesive energy at room temperature, making it difficult to react with hydrogen [44]. Therefore, the hydrogen sensing response at room temperature is very low. With UV illumination, electron-hole pairs are generated in ZnO, and holes react with O_2_^−^ (ad) to produce oxygen gas molecules. Additionally, the gas molecules are ionized again by reacting with photogenerated electrons (O_2_^−^(hν)). These photoinduced ionized oxygen species have weak adhesive energy enabling hydrogen reaction at room temperature [53]. Removing oxygen ion species from the surface acts as a positive surface potential of the AlGaN/GaN FET sensor, which increases the sensor current. Therefore, hydrogen sensing is possible at room temperature with UV illumination. This effect is boosted in ZnO-NPs because of their large surface-to-volume ratios resulted from the small size of ZnO-NPs. The increased standby current with UV illumination is due to the removed O_2_^−^(ad) from the surface.

The sensing response characteristics are defined by [54]:(1)$$\mathrm{Response}\;\left[\%\right]=\left(\frac{I_{gas}-I_{air}}{I_{air}}\right)\times 100,$$
where *I_gas_* is the sensor current with 4% hydrogen injection, and *I_air_* is the sensor current without hydrogen injection. The extracted sensing response characteristics without and with UV illumination are shown in Figure 6c,d, respectively. Remarkable enhancement in the sensing response was observed with UV illumination. A response of ~25% was observed at a bias voltage of 5 V.

The sensing repeatability and time transient characteristics were examined at room temperature using a bias voltage of 5 V. Hydrogen gas with 4% concentration was injected for 20 s and paused for the subsequent 40 s, and this process was repeated. As shown in Figure 8a, stable operations with good repeatability characteristics are observed for both the cases with and without UV illumination. The magnified time transient characteristics are shown in Figure 8b, where the response and recovery times with UV illumination are 8 s and 11 s, respectively, whereas those without UV illumination are 12 s and 18 s, respectively. The response and recovery times were defined as the durations required for the response current to reach 90% and 10% of the saturation current, respectively.

The hydrogen concentration dependent response characteristics were also investigated at room temperature with UV illumination during which the hydrogen concentration was varied from 0.1% to 4%. The hydrogen gas was injected for 20 s at each concentration and the sensor was biased at 5 V. As shown in Figure 9, the sensing current exhibited strong dependency on hydrogen concentration over the entire range; the current increases with increasing hydrogen concentration.

In Table 1, the sensor characteristics are compared with other hydrogen sensors reported at room temperature. The hydrogen sensor fabricated in this work exhibited very fast response and recovery characteristics with a wide detection range. It is suggested that the sensing response can be further improved by employing a thinner AlGaN barrier layer that can reduce the standby current (*I_air_* in Equation (1)) level [55].

## 3. Conclusions

UV-assisted photocatalytic hydrogen sensing capability was demonstrated at room temperature using an AlGaN/GaN based sensor with a dual catalyst layer of ZnO-NP/Pd. A sensing response of 25% with a response time of 8 s and a recovery time of 11 s was achieved for a hydrogen concentration of 4% at room temperature under 280 nm UV illumination. The room temperature operation can thus eliminate the process of heating that is generally required for hydrogen catalytic reactions using conventional materials. Therefore, the proposed sensor has advantages of less power consumption and no need for stabilization. To the best of our knowledge, this is the first demonstration of the operation of a UV-assisted AlGaN/GaN hydrogen sensor at room temperature.

## Figures and Tables

**Figure 1 nanomaterials-11-01422-f001:**
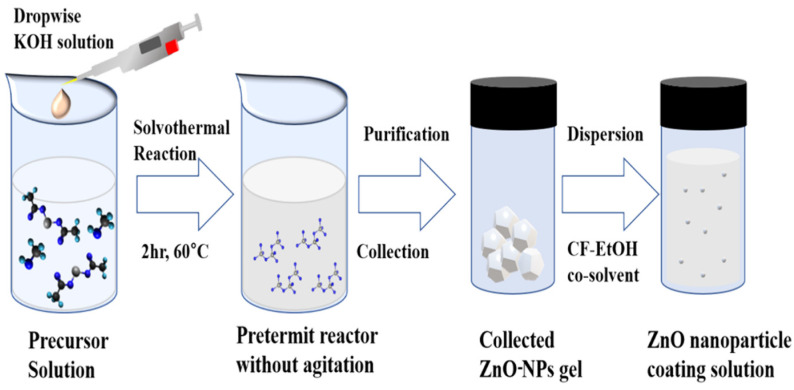
Synthesis process for ZnO-NPs.

**Figure 2 nanomaterials-11-01422-f002:**
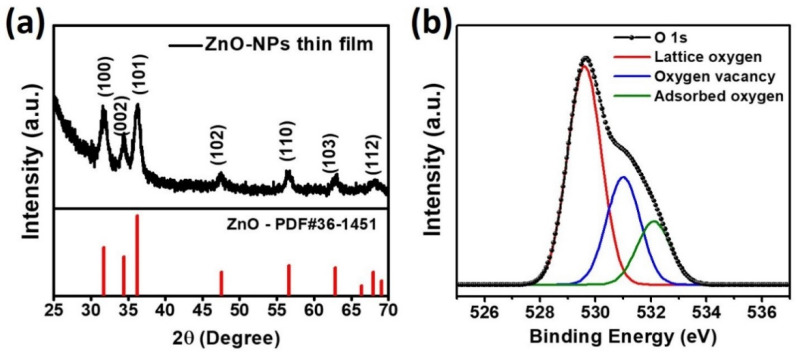
(**a**) X-ray diffraction pattern and (**b**) O 1s XPS spectra of as-synthesized ZnO-NPs.

**Figure 3 nanomaterials-11-01422-f003:**
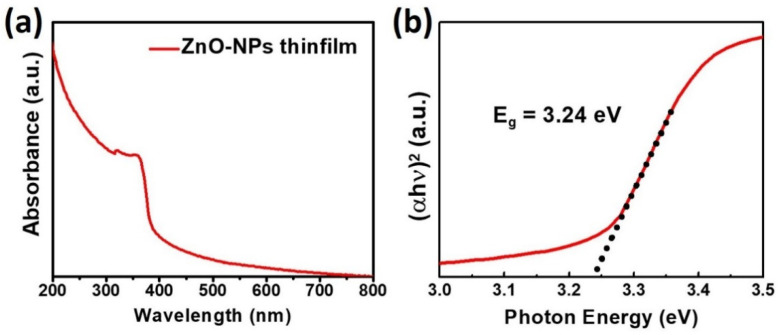
(**a**) Absorption spectrum and (**b**) Tauc plot of ZnO-NPs thin film.

**Figure 4 nanomaterials-11-01422-f004:**
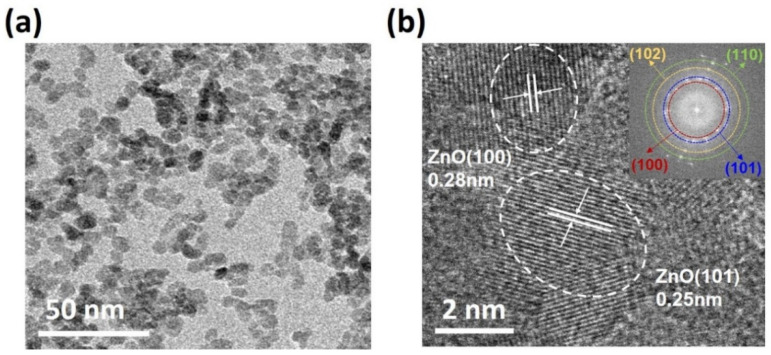
(**a**) TEM and (**b**) HRTEM image of as-synthesized ZnO NPs (inset: FFT pattern).

**Figure 5 nanomaterials-11-01422-f005:**
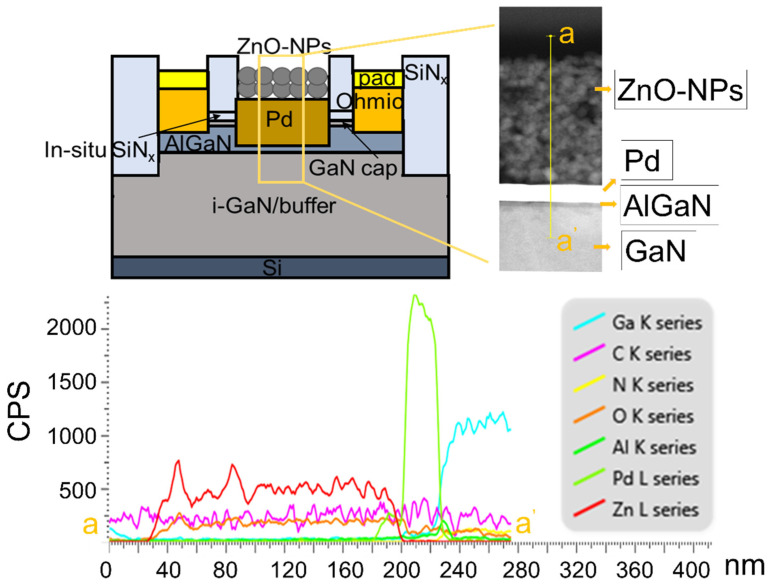
Cross-sectional schematic of AlGaN/GaN based hydrogen sensor with a dual catalyst layer. The inset shows the cross-sectional TEM image of the dual catalyst layer of ZnO-NP/Pd.

**Figure 6 nanomaterials-11-01422-f006:**
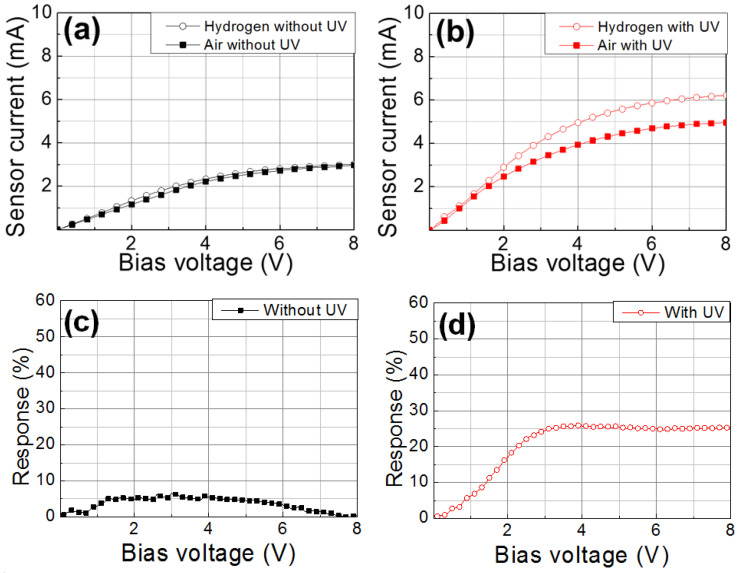
Sensing characteristics of a fabricated AlGaN/GaN based hydrogen sensor with ZnO-NP/Pd dual catalyst layer at room temperature without and with UV illumination; current–voltage characteristics (**a**) without and (**b**) with UV illumination and the corresponding response characteristics (**c**) without and (**d**) with UV illumination.

**Figure 7 nanomaterials-11-01422-f007:**
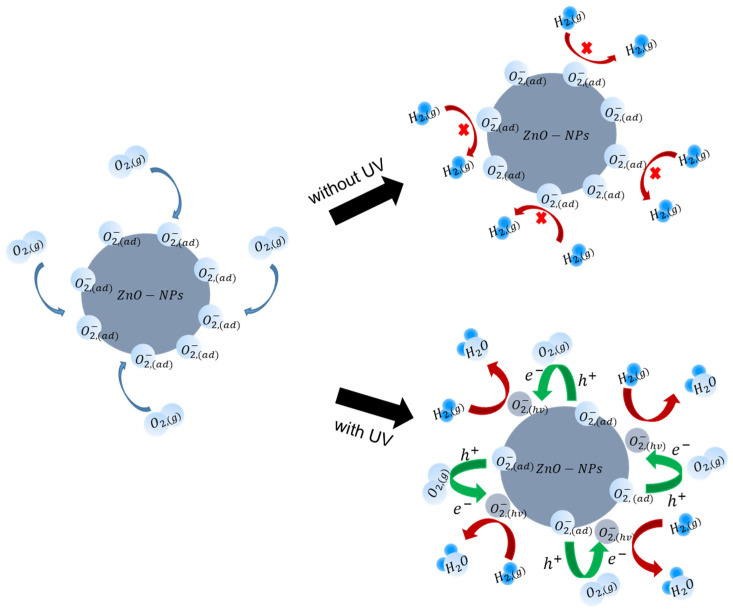
Schematic illustration of UV photocatalytic reaction of hydrogen with ZnO-NP.

**Figure 8 nanomaterials-11-01422-f008:**
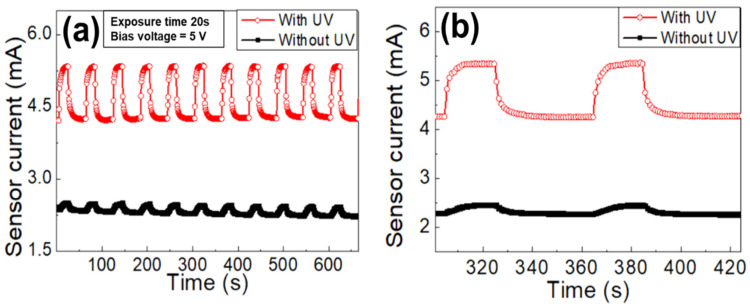
(**a**) Sensing repeatability characteristics of AlGaN/GaN hydrogen sensor with and without UV illumination and (**b**) zoom-in time transient characteristics.

**Figure 9 nanomaterials-11-01422-f009:**
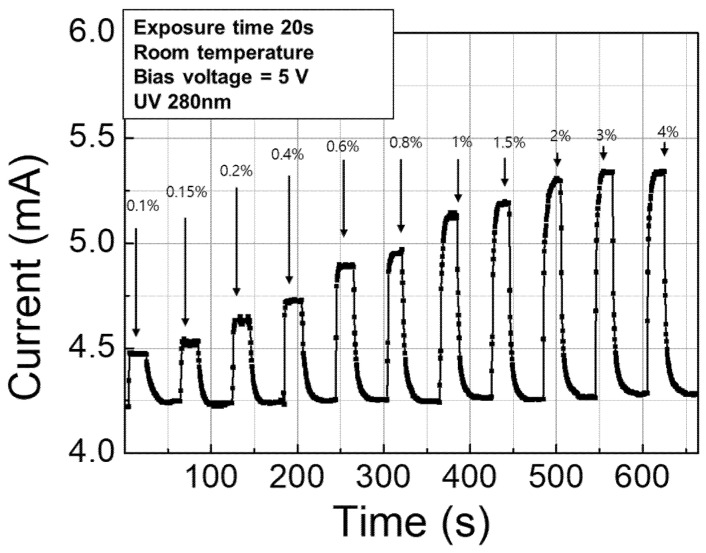
Hydrogen concentration dependent response characteristics of AlGaN/GaN hydrogen sensor at room temperature with UV illumination.

**Table 1 nanomaterials-11-01422-t001:** Comparison of hydrogen sensor characteristics reported at room temperature.

Sensor Type	Catalyst	Catalyst Structure	Hydrogen Concentration	Response Time	Recovery Time	Sensor Response	Ref
Resistive	Au/ZnO	Nanoparticle	0.0005%	4 s	24 s	21.5%	[44]
Resistive	ZnO	Nanoline	0.01%	~12 min	~20 min	19%	[53]
Resistive	Pd	Nanowire	0.15%	~10 min	~10 min	9.1%	[56]
Resistive	Pt	Nanoparticle/nanowire	0.2%	-	-	62%	[57]
Resistive	Pd	Nanoparticle/nanofiber	0.1%	~6 s	~3 s	12.09%	[58]
Resistive	Pt/SnO_2_	Nanoparticle/nanoparticle	0.1%	~20 s	~80 s	10,500%	[59]
Resistive	ZnO	Nanorod	0.05%	~15 min	~20 s	4.2%	[60]
Resistive	W_18_O_4_	Nanowire	0.0002%	-	-	~1%	[61]
Resistive	VO_2_	Nanobelts	0.014%	~840 s	~455 s	~1800%	[62]
Resistive	SnO_2_	Nanoparticle	0.1%	205 s	116 s	600%	[63]
Resistive	SnO_2_	Nanobelts	2%	~220 s	~220 s	50%	[64]
Resistive	Pd	Nanoparticle/nanotube	1%	2 min	1.5 min	9.5%	[65]
Diode (AlGaAs MOS)	Pd	Thin film	1%	58 s	-	155.9%	[66]
Diode (AlGaAs Schottky)	Pd	Thin film	1%	400 s	-	~5%	
Diode (GaN Schottky)	Pt	Thin film	1%	15 s	19 s	1 × 10^5^%	[67]
FET (AlGaN/GaN)	ZnO/Pd	Nanoparticle/thin film	4%	8 s	11 s	25%	This work

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
