# Peer review of "Room Temperature Operation of UV Photocatalytic Functionalized AlGaN/GaN Heterostructure Hydrogen Sensor"

_nanomaterials, 2021, doi:10.3390/nano11061422_

Round 1

Reviewer 1 Report

In the presented article, the authors investigated the possibility of using UV radiation to activate the processes of hydrogen detection at room temperature by a detector based on a AlGaN/GaN heterostructure. Catalyst layer contained ZnO-nanoparticles that covered a Pd film. The device (architecture) of the detector, the main characteristics of the functional layers of this detector and methods for their creation were previously described by the authors in the article “AlGaN/GaN heterojunction hydrogen sensor using ZnO-nanoparticles/Pd dual catalyst layer” June-Heang Choi, Taehyun Park, Jaehyun Hur, and Ho-Young Cha, Sensors & Actuators: B. Chemical 325 (2020) 128946. Some of the results have been transferred virtually unchanged to the submitted manuscript. This is largely necessary for understanding the main characterization of the detector, but the Communication should have focused on new important information. The authors added the results of studying the optical properties of ZnO nanoparticles and measured the detector's response to hydrogen at room temperature. However, several issues require clarification.

In the introduction, it is desirable to substantiate the purpose of the work in more detail and describe the prerequisites for using UV radiation for hydrogen detection by AlGaN/GaN heterojunction hydrogen sensor using ZnO-nanoparticles/Pd dual catalyst layer.

There is no data on the chemical state of the surface of ZnO nanoparticles, which can have an important effect on surface reactions under the influence of UV radiation. It is desirable to provide data on the effect of hydrogen concentration on the detector response.

Please, conduct a more detailed analysis of the processes occurring in the catalytic layers under the influence of hydrogen and UV radiation. It is desirable to add a schematic illustration of these processes.

The authors stated that the created detector had “excellent response characteristics” in the case of hydrogen detection at room temperature. It is necessary to substantiate this statement and compare the obtained characteristics with the response characteristics of other detectors (for example, resistive detectors based on Pd/ZnO-nanorods, ZnO/WO3 core-shell nanowires, etc.).

Please check the authors affiliations: 1 - Korea (?); 2 - Republic(?)

Author Response

Thank you for the kind suggestion. 

We have revised the paper based on the amendments you suggested.

Thank you.

Best regard

Reviewer 2 Report

In this paper, an AlGaN/GaN heterostructure based hydrogen sensor was fabricated using a dual catalyst layer with ZnO-nanoparticles (NPs) atop of Pd catalyst film. The work is quite innovative, and the following problems need to be solved.

  1. In the introduction part, the author should mention the application of zinc oxide and other semiconductors as catalysts in the field of Hydrogen Sensor.
  2. In the experimental part, the author mentioned the ZnO particle film, and how thick is the film?
  3. In the main part, in order to express the advantages of the device more intuitively, the author should make a table of performance comparison of related devices.
  4. ZnO particles play a very important role in the device, and the author needs to elaborate more from the physical mechanism.

Author Response

(The authors gave the same response as above.)

Round 2

Reviewer 1 Report

The authors retained the claim that they received “excellent response characteristics” in the case of hydrogen detection at room temperature.  However, analyzing the comparison results (Table 1) they revealed relatively lower response in comparison with Si FET sensor and proposed that the response characteristics can be further improved. These facts raise doubts that the response characteristics of the created detectors are "excellent". A different definition should be chosen for the achieved performance.

Author Response

(The authors gave the same response as above.)

Reviewer 2 Report

The article has been modified systematically and can be accepted.

Author Response

Thank you for accepting.

Thank you.

Best regard
